# Six-month mortality has decreased for patients with curative treatment intent for head and neck cancer in Sweden

Charbél Talani[1,2]*, Anders Högmo[3], Göran Laurell[4], Antti Mäkitie[5,6,7], Lovisa Farnebo[1,2]

1 Division of Sensory Organs and Communication, Department of Biomedical and Clinical Sciences, Faculty of Medicine and Health Sciences, Linköping University, Linköping, Sweden, 2 Department of Otorhinolaryngology, Region Östergötland Anesthetics, Operations, and Specialty Surgery Center, Linköping, Sweden, 3 Regional Cancer Center Western Sweden, Sahlgrenska University Hospital, Gothenburg, Sweden, 4 Department of Surgical Sciences, Uppsala University, Uppsala, Sweden, 5 Department of Otorhinolaryngology—Head and Neck Surgery, University of Helsinki and Helsinki University Hospital, Helsinki, Finland, 6 Division of Ear, Nose and Throat Diseases, Department of Clinical Sciences, Intervention and Technology, Karolinska Institutet and Karolinska Hospital, Stockholm, Sweden, 7 Research Program in Systems Oncology, Faculty of Medicine, University of Helsinki, Helsinki, Finland

* Charbel.talani@regionostergotland.se

**Data Availability Statement:** All data are owned by a third party organization. The Swedish Head and Neck cancer Registry (SweHNCR) and Regional Cancer Center Väst owns the data. The SweHNCR

## Abstract

### Background

In general, survival outcomes for patients with Head and Neck Cancer (HNC) has improved over recent decades. However, mortality within six months after diagnosis for curative patients remains at approximately 5%. The aim of this study was to identify risk factors for early death among patients with curative treatment, and furthermore, to analyze whether the risk of early death changed over recent years.

### Material and method

This real-world, population-based, nationwide study from the Swedish Head and Neck Cancer Register (SweHNCR) included all patients ≥18 years diagnosed with HNC with a curative treatment intent at the multidisciplinary tumor board from 2008 to 2020. A total of 16,786 patients were included.

### Results

During the study period a total of 618 (3.7%) patients with curative-intended treatment died within six months of diagnosis. Patients diagnosed between 2008 and 2012 had a six-month mortality rate of 4.7% compared to 2.5% for patients diagnosed between 2017 and 2020, indicating a risk reduction of 53% (p <0.001) for death within six months. The mean time to radiation therapy from diagnosis in the 2008–2012 cohort was 38 days, compared to 22 days for the 2017–2020 cohort, (p <0.001). The mean time to surgery from diagnosis was 22 days in 2008–2012, compared to 15 days for the 2017–2020 cohort, (p <0.001). Females had a 20% lower risk of dying within six months compared to males (p = 0.013). For every

contains data on almost 99% of all Head and Neck cancer patients in Sweden. The registry started in 2008. The Registry contains data on sex, age, TNM score, WHO score, smoking habits, date of diagnosis, date of death, stage, tumor site, treatment given, treatment intent, date of treatment, time from diagnosis to treatment. The authors did not receive any special privileges accessing the data that other researchers would not have. To access the data please file a formal application via this link. https://cancercentrum.se/samverkan/vara-uppdrag/kunskapsstyrning/kvalitetsregister/datauttag/ Questions regarding the Registry and access to data can be e-mailed to: arvid.widenlou.nordmark@regionvasterbotten.se.

**Funding:** This study was supported in the form of funding by Region Östergötland (Grant No. SC-2018-00231-53) awarded to CT, Region Östergötland (Grant No. SC-2018-00231-46) awarded to LF, Cancerfonden (Grant No. 2018/502) awarded to GL, and Finska Läkaresällskapet (Grant No. AM2023) awarded to AM.

**Competing interests:** The authors have declared that no competing interests exist.

year older the patient was at diagnosis, a 4.8% (p <0.001) higher risk of dying within six months was observed. Patients with a WHO score of 1 had approximately 2.4-times greater risk of early death compared to WHO 0 patients (p <0.001). The risk of early death among WHO 4 patients was almost 28 times higher than for WHO 0 patients (p <0.001). Patients with a hypopharyngeal tumor site had a 2.5-fold higher risk of dying within six months from diagnosis compared to oropharyngeal tumor patients (p <0.001).

## Conclusions

We found that the risk of early death decreased significantly from 2008 to 2020. During this period, the mean time to the start of treatment was significantly reduced both for surgery and oncological treatment regimes. Among patients with a curative treatment intention, increased risk of early death was associated with male sex, older age, advanced disease, increased WHO score, and a hypopharyngeal tumor site.

## Background

Head and neck cancer (HNC) is one of the most common cancers worldwide, and epidemiological assessments have reported a recent global increasing incidence [1–5], in Sweden as well as elsewhere in the recent years [6, 7]. Simultaneously, a reduced mortality has been noted in HNC in recent decades, due to better available treatments, less smoking, a multidisciplinary approach to treatment, and earlier discovery of tumors [8–11].

Numerous studies have dealt with survival in patients with HNC, but population-based studies focusing on early death in curatively treated patients with HNC are scarce [12, 13]. Early death from HNC was previously defined as patients dying within six months of diagnosis in studies from the Swedish Head and Neck Cancer register (SweHNCR) [12, 14]. 4.5% of patients died within six months of diagnosis between 2008 and 2015, even though they received curative-intended treatment [12]. Curative-intended treatment is often aggressive, costly, and comes with considerable side effects [15, 16]. If patients receiving aggressive treatment die within six months, the physicians involved must address whether the indications and decision-making involved in treatment management should be re-evaluated [17]. Both surgery and radiotherapy—the backbones of head and neck cancer treatment [5, 17, 18]—warrant repeated visits to the hospital and frequent hospitalizations. Understanding the variables linked to patients at risk of early death could help clinicians better select suitable treatments, and thus avoid mutilating surgery and aggressive chemoradiotherapy in cases with a considerable risk of early death.

Sweden has a single-payer healthcare system. Major progress in clinical management of patients with HNC has been made in Sweden since 2008. Care has, in general, been influenced by international initiatives aiming to improve quality of care of patients with HNC [5, 7]. The Swedish National Guidelines for HNC were established 2015 and have since then been used at the tertiary referral hospitals in the country [18]. Therefore, to study the efficacy of the new guidelines in an era of Human Papilloma Virus (HPV) and demographic changes, further analysis of patients dying within six months of treatment is warranted. In this large, real-world, nationwide, population-based study containing 18,739 patients with HNC (16,786 curative and 1769 palliative), the primary aim was to identify patients at risk of early death after treatment with curative intent, and to analyze factors that contribute to this increased risk.

Furthermore, we wanted to test the hypothesis as to whether six-month mortality has decreased for HNC patients over recent years.

## Material and methods

### Data source

Data were obtained from the SweHNCR (Ethics Committee approval; number 2020–01972). The SweHNCR is funded by the Swedish government and includes 98.5% of all HNC patients since 2008 when cross referenced with the register of the Department of Health and Welfare in Sweden. The authors did not have access to information that could identify individual participants during or after data collection. Application for data extraction was sent to Regional Cancer Centre Väst 15 October 2021. Decision of disclosure was made 12 November 2021, and the data file was delivered on November 26, 2021 (Number: SV2803, Diary number: HS2021-01178). Data reported to the SweHNCR included: diagnosis, TNM classification (according TNM 7 [19]), stage, sex, age, WHO score at diagnosis, time to treatment, treatment, follow-up, recurrence, and survival.

The total number of consecutively-affected Swedish patients during the period of January 1, 2008–December 31, 2020 in SweHNCR with at least six months of follow-up was 18,739 [5].

### Patient population

All consecutive HNC patients, ≥18 years of age who were diagnosed in Sweden were included in the study. Patients with a palliative treatment decision at multidisciplinary tumor board (MDT) (n = 1953, 10.4%) were excluded, leaving 16,786 patients with curative treatment intent for further analysis. Patients with curatively intended treatment were divided into three groups based on year of diagnosis: 2008–2012 (n = 5778), 2013–2016 (n = 5307), and 2017–2020 (n = 5701) (Fig 1). All three groups had a similar duration, and included a comparable number of patients.

Ten tumor sites (classified according to ICD-10 codes) were included: lip (C00.0–2, C00.6, C00.8, C00.9); oral cavity (C00.3, C00.4, C02, C03, C04, C05, C06); oropharynx (C01.9, C05.1, C05.2, C05.8, C05.9, C09, C10); nasopharynx (C11); hypopharynx (C12, C13); larynx (C10.1, C32); nose (C30.0) and nasal sinuses (C31); salivary glands (C07, C08); head and neck cancer of unknown primary origin (C77.0); and middle or inner ear (C 30.1). Malignant tumors located in the esophagus, thyroid, or parathyroid glands were not included, since information regarding these diagnoses is not included in the SweHNCR. The Eastern Cooperative Oncology Group score—the WHO score—runs from 0 to 5, with 0 denoting perfect health and 5 death.[20] The WHO score was rated 0–4, indicating the physical performance status of the patient. A higher score indicates worse physical performance.

HNC treatment in Sweden is centralized to seven university hospitals, although a few regional hospitals provide non-surgical oncological treatment. National Healthcare and Social Security systems are offered equally to all Swedish inhabitants. All patients are examined by either an oncologist or a head and neck surgeon after termination of treatment to evaluate treatment outcome every three months for the first two years post-treatment and every sixth months for three years after that [18].

### Statistics

Results are presented as the mean, standard deviation, and range for continuous variables, and as numbers and percentages for categorical variables. Incidence rates of HNC for 2008–2020 and incidence rate ratio was calculated comparing 2020 to 2008 with 95% confidence intervals

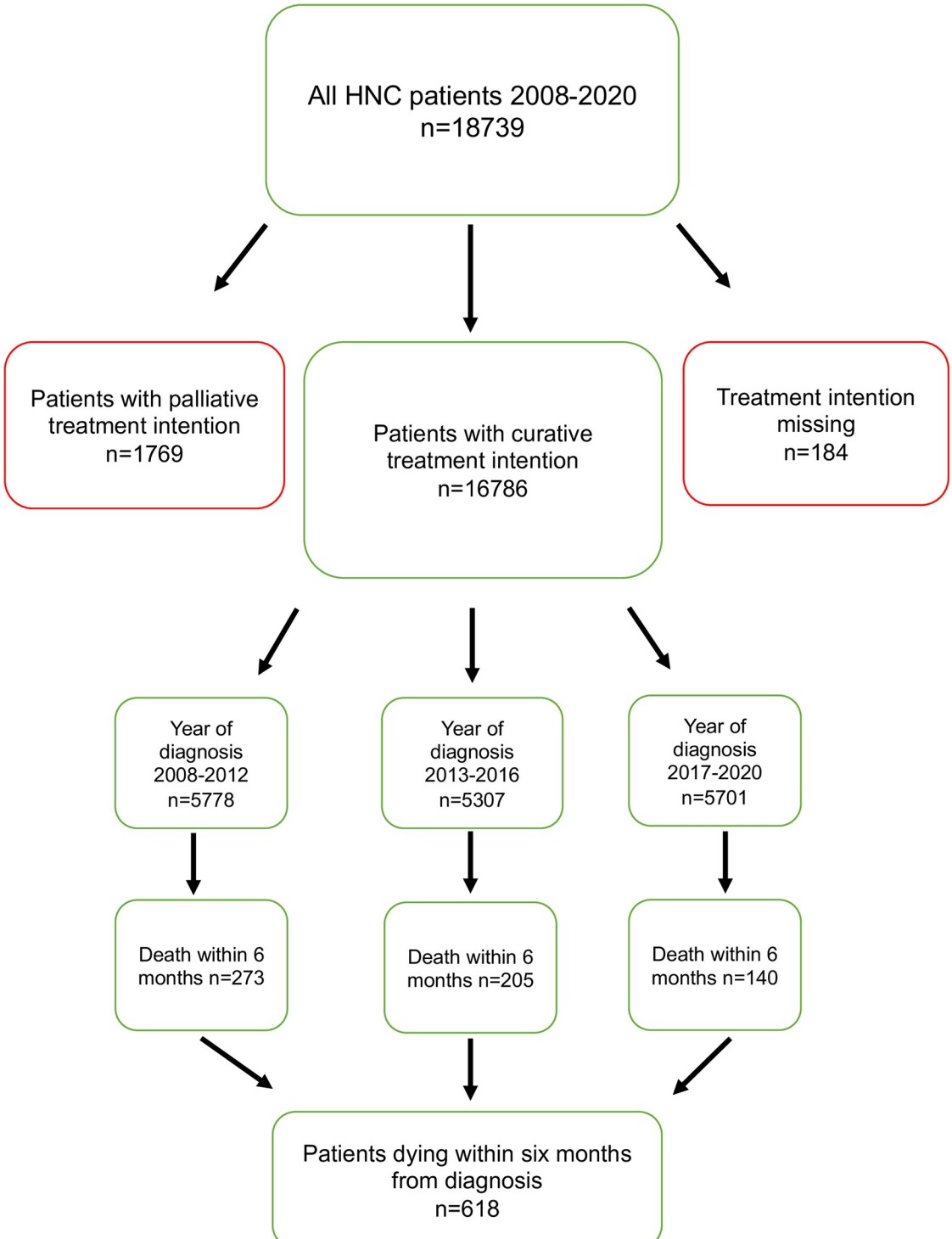

**Fig 1. Flow chart describing treatment intent, year of diagnosis, and early death for included patients in green boxes.** Excluded patients are shown in red boxes.

according to a method described by Martin and Austin [6, 21–23]. Data from Statistics Sweden were used for information regarding yearly Swedish population count. SweHNCR was used to identify the number of new HNC cases annually. For comparisons between two groups (divided by: age, days to treatment, and days from diagnosis to death), the independent student's t-test was used. ANOVA tests (with Bonferroni method as a post hoc test) and chi-square tests were used for categorical variables (stage, site, age group, sex, TNM score, death, time periods, and WHO function class). Exact binomial confidence intervals were estimated for proportions. Logistic regression analysis was used to control for confounding factors. To describe overall crude survival and early death for subgroups, a Kaplan—Meier plot was used, and the difference between subgroups (sex, time periods, age groups, and WHO class) was analyzed with a log-rank test. Cox regression was used for multivariable analyses with death within six months from diagnosis as failure. All significance tests were two-tailed and conducted at a 5% significance level. For all statistical analyses, we used IBM SPSS Statistics for Macintosh, Version 28.0. Armonk, NY: IBM Corp Released 2021.

## Results

### Epidemiology

The annual number of new patients with HNC steadily increased from 2008 (n = 1211 cases) to 2020 (n = 1673 cases). During the same period, the Swedish population increased by 12%, from 9.2M to 10.4M, indicating that factors other than a growing population were involved in the increase of new HNC cases. We noted an incidence rate ratio of 23.8% (CI 95% 23.5–24.2) between 2008 (13.0 diagnosed with HNC per 100000) and 2020 (16.1 diagnosed with HNC per 100000), corresponding to an annual incidence increase of 0.25 per 100000 (p <0.001) (calculated as described in the statistics section above) (Fig 2). Both oropharyngeal and oral cavity cancer displayed increased occurrence: the number of patients with oropharyngeal cancer rose 86% from 2008 (n = 270) to 2020 (n = 503); oral cancer rose 48% from 2008 (n = 343) to 2020 (n = 509). The incidence rate ratio for oropharyngeal cancer was 66.1% (CI 95% 64.3–68.0) between 2008 (2.92 per 100000) and 2020 (4.85 per 100000), and for oral cavity cancer was 32.3% (CI 95% 32.0–33.6) from 2008 (3.71 per 100000) to 2020 (4.90 per 100000). All other sites had fairly similar occurrence rates over the years 2008–2020 (Fig 2). The number of patients discussed at multidisciplinary tumor boards (MDT) increased from 2008 to 2020 (95% to 99.5%, mean for the total study period of 97%).

During the years 2008–2020, 618 (3.7%) of 16,786 patients died within six months of diagnosis, despite being recommended for curatively intended treatment at MDT (Fig 1).

### Predictors for six-month mortality

**Year of diagnosis.** The total cohort was divided into three groups based on the year of diagnosis. The three groups contained a comparable number of patients: 2008–2012 (n = 5778), 2013–2016 (n = 5307), and 2017–2020 (n = 5701) (Tables 1 and 2). Between 2008–2012, six-month mortality was 4.7% compared to 3.9% and 2.5% for the years 2013–2016 and 2017–2020, respectively (p <0.001) (Table 1, Fig 3). Sex, age, and stage were evenly distributed throughout the three groups. The 2008–2012 group had a slightly higher distribution of unknown WHO scores compared to the two latter groups, (p <0.001) (Table 2). The composition of oropharyngeal and oral cavity cancer differed between the 2017–2020 cohort (30.5% and 28.3% of all HNC cancers, respectively) compared to the 2008–2012 cohort (25.3% and 26.3% respectively), (p <0.001).

**Time to treatment.** Time to start of treatment from diagnosis was shorter in the 2017–2020 group compared to the 2008–2012 group. The mean time to radiotherapy/

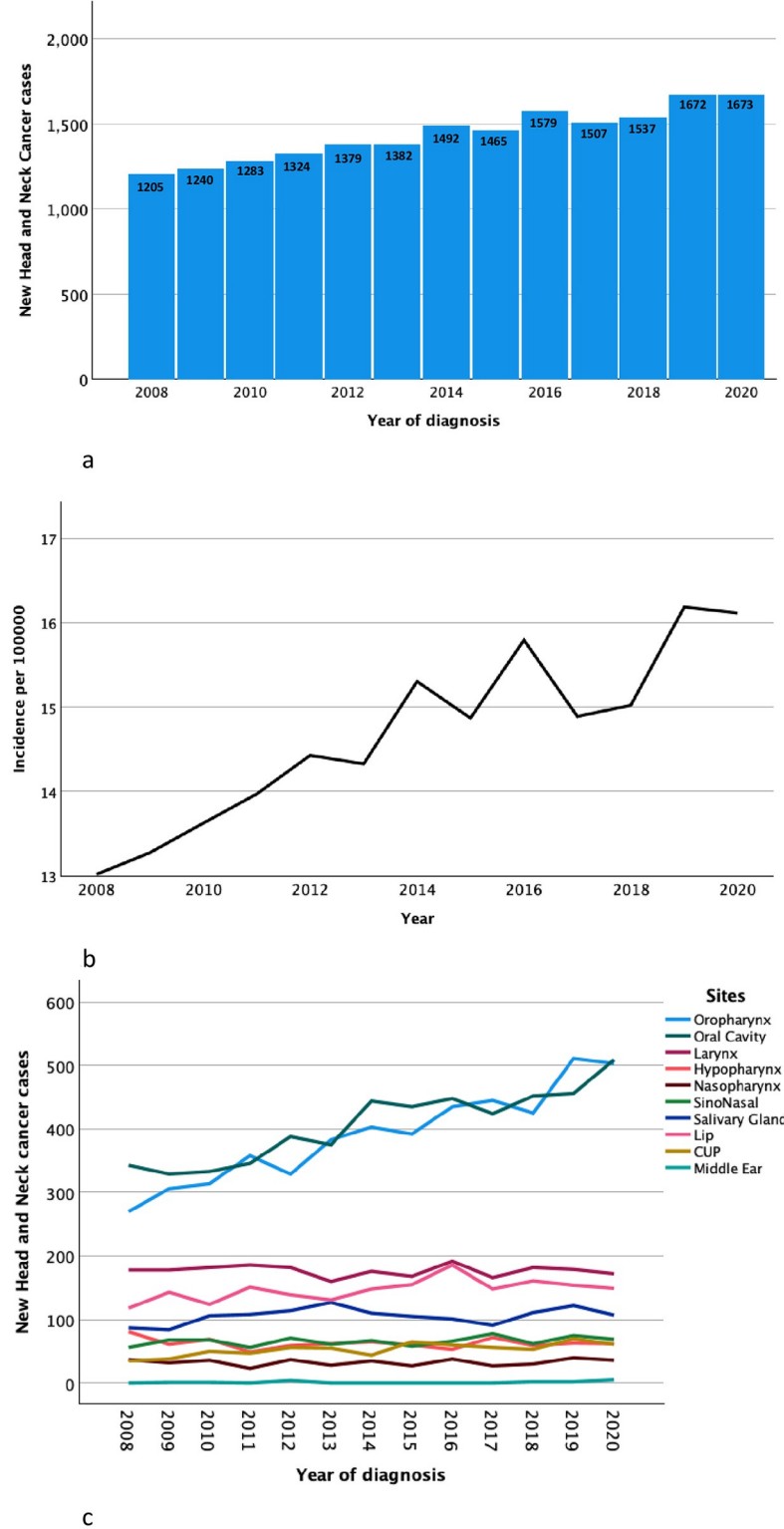

**Fig 2.** a. Annual number of new cases of head and neck cancer in Sweden per year 2008–2020. b. Incidence of head and neck cancer in Sweden 2008–2020. c. Distribution of tumor sites of head and neck cancer in Sweden 2008–2020.

**Table 1. Descriptive data for patients with curative treatment intent, and univariate analysis of death within 6 months of diagnosis.**

| N = 16786 | Prevalence n (%) | Dead within 6 months/N (%) | Hazard Ratio (95% CI) | P-values |
|---|---|---|---|---|
| **Sex** | | | | |
| Male | 10,712 (63.8) | 424/10,712 (4.0) | 1.00 | - |
| Female | 6074 (36.2) | 194/6074 (3.2) | 0.80 (0.68–0.95) | 0.012 |
| **Total** | 16,786(100) | 618/16,786(3.7) | - | - |
| **Age** (mean (sd)) | 66.1 (12.6) | | | |
| **Age group in years** | | | | |
| 18–39 | 463 (2.8) | 0/463 (0) | - | - |
| 40–49 | 1139 (6.8) | 14/1139 (1.2) | 1.00 | - |
| 50–59 | 3043 (18.1) | 52/3043 (1.7) | 1.40 (0.77–2.52) | 0.268 |
| 60–69 | 5378 (32.0) | 155/5378 (2.9) | 2.37 (1.37–4.10) | 0.002 |
| 70–79 | 4368 (26.0) | 203/4368 (4.6) | 3.86 (2.24–6.62) | <0.001 |
| 80+ | 2395 (14.3) | 194/2395 (8.1) | 6.88 (4.00–11.8) | <0.001 |
| **WHO Score** | | | | |
| 0 | 12,078 (72.0) | 206/12,078 (1.7) | 1.00 | - |
| 1 | 2244 (13.4) | 145/2244 (6.5) | 3.88 (3.14–4.80) | <0.001 |
| 2 | 890 (5.3) | 108/890 (12.1) | 7.53 (5.97–9.51) | <0.001 |
| 3 | 321 (1.9) | 68/321 (21.2) | 14.4 (11.0–19.0) | <0.001 |
| 4 | 36 (0.2) | 16/36 (44.4) | 37.2 (22.4–62.0) | <0.001 |
| Missing | 1217 (7.3) | 75/1217 (12.1) | 3.72 (2.85–4.84) | <0.001 |
| **Year of diagnosis** | 16,786 (100) | 618/16,786 (3.7) | - | - |
| 2008–2012 | 5778 (34.4) | 273/5778 (4.7) | 1.00 | - |
| 2013–2016 | 5307 (31.6) | 205/5307 (3.9) | 0.82 (0.68–0.98) | 0.026 |
| 2017–2020 | 5701 (34.0) | 140/5561 (2.5) | 0.51 (0.42–0.63) | <0.001 |

chemoradiation therapy in the 2008–2012 cohort was 38 days, compared to 22 days for the 2017–2020 cohort, p <0.001. The mean time to surgery from diagnosis was 22 days in 2008–2012, compared to 15 days for the 2017–2020 cohort, p <0.001 (Table 2).

**Sex.** Altogether, 10,712 (63.8%) of patients were male, and 6074 (36.2%) were female (Table 1). A significant difference in mortality between the sexes was found. The six-month mortality rate among male patients was 4.0% compared to 3.2% for females (p = 0.012) (Table 1) (Fig 4). Therefore, female patients had a 20% lower risk of death within six months compared to male patients.

**Age.** The mean age at diagnosis was 66 years (range: 18–102). Males were on average one year younger than the females (65.7 versus 66.7) at diagnosis (p <0.001). No patient below 40 years of age died within six months, compared to 8.1% of all patients older than 80 years (p <0.001). There was a 6.88-fold increased risk of dying within six months for patients >80 years of age compared to patients <50 years of age (Table 1) (Fig 4).

**WHO score.** 12,078 (72%) of all patients had a WHO score of 0. Only 1.7% of all patients with WHO 0 died within six months, compared to 44.4% of those with WHO 4 (p <0.001). In univariate analysis, the risk of death within six months was 37.2 times higher for patients with WHO 4 compared to those with WHO 0 (Table 1). Mean survival time among patients dying within 6 months was 119 days for WHO 0 patients compared to 76 days for WHO 4 patients (p = 0.004) (Fig 4).

**Site.** The most common tumor site was the oropharynx (28%), followed by the oral cavity (27.8%) and larynx (12.7%) (Table 3). The six-month mortality rate varied between tumor

**Table 2. Descriptive data for patients with curative treatment intent grouped by time of diagnosis and univariate analyses.**

| Variable | 2008–2012 | 2013–2016 | 2017–2020 | Sig. (p-value) |
|---|---|---|---|---|
| Total number of patients N (%) | 5778 (100) | 5397 (100) | 5791 (100) | - |
| Oropharynx n (%) | 1461 (25.3) | 1499 (28.2) | 1738 (30.5) | <0.001 |
| Oral cavity n (%) | 1531 (26.5) | 1514 (28.5) | 1615 (28.3) | <0.001 |
| Other sites n (%) | 2786 (48.2) | 2294 (43.2) | 2348 (41.2) | <0.001 |
| Male n (%) | 3713 (64.3) | 3416 (64.4) | 3583 (62.8) | 0.174 |
| Female n (%) | 2065 (35.7) | 1891 (35.6) | 2118 (37.2) | 0.173 |
| Age n (%) 18–39 | 183 (3.2) | 138 (2.6) | 142 (2.5) | 0.337[a] |
| Age n (%) 40–49 | 410 (7.1) | 376 (7.1) | 353 (6.2) | - |
| Age n (%) 50–59 | 1113 (19.3) | 964 (18.2) | 966 (16.9) | 0.710[a] |
| Age n (%) 60–69 | 1999 (34.6) | 1696 (32.0) | 1683 (29.5) | 0.600[a] |
| Age n (%) 70–79 | 1247 (21.6) | 1358 (25.6) | 1763 (30.9) | <0.001[a] |
| Age n (%) 80+ | 826 (14.3) | 775 (14.6) | 794 (13.9) | 0.425[a] |
| Stage I n (%) | 1722 (31.1) | 1538 (30.4) | 1688 (31.1) | - |
| Stage II n (%) | 1063 (19.2) | 930 (18.4) | 854 (15.8) | <0.001[b] |
| Stage III n (%) | 784 (14.2) | 617 (12.2) | 676 (12.5) | 0.063[b] |
| Stage IV n (%) | 1967 (35.5) | 1982 (39.1) | 2204 (40.6) | 0.007[b] |
| WHO 0 n (%) | 4005 (69.3) | 3780 (71.2) | 4293 (75.3) | - |
| WHO 1 n (%) | 720 (12.5) | 763 (14.4) | 761 (13.3) | 0.040[c] |
| WHO 2 n (%) | 289 (5.0) | 287 (5.4) | 314 (5.5) | 0.830[c] |
| WHO 3 n (%) | 108 (1.9) | 95 (1.8) | 118 (2.1) | 0.802[c] |
| WHO 4 n (%) | 14 (0.2) | 12 (0.2) | 10 (0.2) | 0.603[c] |
| WHO missing n (%) | 642 (11.1) | 370 (7.0) | 205 (3.6) | <0.001[c] |
| No treatment n (%) | 39 (0.7) | 27 (0.5) | 29 (0.5) | 0.697[d] |
| Radiotherapy +/- Chemotherapy n (%) | 2488 (43.9) | 2382 (45.6) | 2608 (46.3) | <0.001[d] |
| Single modality surgery n (%) | 1703 (30.1) | 1664 (31.8) | 2011 (35.7) | <0.001[d] |
| Combination of surgery + radiotherapy +/- chemotherapy n (%) | 1432 (25.3) | 1152 (22) | 987 (17.5) | - |
| Time to surgery, mean days** | 22 | 19 | 14 | <0.001 |
| Time to radiotherapy, mean days** | 38 | 30 | 22 | <0.001 |

[a] Chi-square between age group 40–49 and other age groups

[b] Chi-square between Stage I and other Stages

[c] Chi-square between WHO 0 and other WHO scores

[d] Chi-square between combined modality treatments and other treatments

**Mean duration in days between 2008–2020

sites. Worst outcomes were seen in patients with hypopharyngeal cancer, with 10.3% dying within six months, despite curative treatment intent. A univariable analysis showed that patients with a hypopharyngeal tumor had an almost 2.5-fold higher relative risk of dying within six months compared to oropharyngeal patients (p <0.001) (Table 3).

**Stage.** An association between tumor stage and six-month mortality was found. Among all patients in the total cohort, 54% were diagnosed with stage III or stage IV cancer (Table 3), although 55% of all women were diagnosed in stage I–II compared to 45% of all males (p <0.001). Of the 618 patients who died within 6 months, 76% had stage III or IV disease compared to 12% of patients with stage I. Significant differences in six-month mortality were also noted in stage IV. For stage I patients, six-month mortality was 1.5%, compared to 5.3%, 10.1%, and 15.1% for stages IV A, IV B, and IV C, respectively (p <0.001). In total, a patient

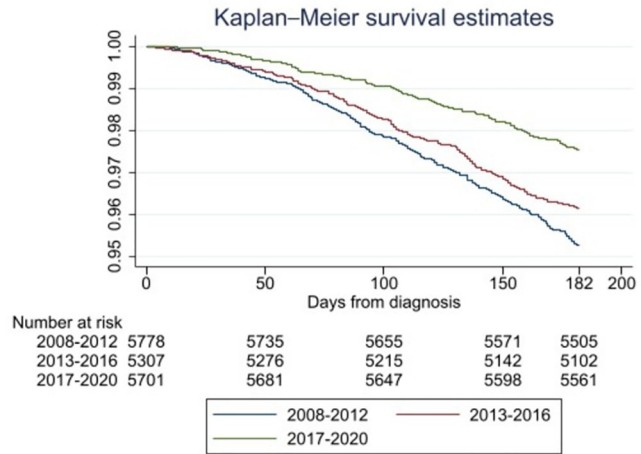

**Fig 3. Kaplan–Meier curve of death within six months from diagnosis based on year of diagnosis, log rank test (p <0.001).**

with stage IV C disease had a 10-times higher risk of dying within six months compared to stage I patients (Table 3).

**TNM class.**   Higher T class correlated with worse prognosis. Seventy percent of all patients had a T1-2 tumor at diagnosis (Table 3). Only 2% of the patients with a T1-2 tumor died within six months compared to 7.7% of T3-4 patients. We found that a patient with T3-4 tumor had a 3 times higher risk of death within six months compared to T1-2 patients (p <0.001) (Table 3).

A total of 11,649 (61%) of all patients had no neck metastasis (N–) at diagnosis, whereas 6520 (39%) had one or more metastases in the neck at diagnosis (N+). The six-month mortality for N—patients was 2.9%, compared to 4.9% for N+ patients (p <0.001). In total, an N + patient had a 1.68-fold higher risk of dying within six months compared to a N—patient (Table 3).

A total of 102 (0.6%) patients had distant metastasis (M1) and were still considered (at the MDT) to benefit from curative treatment. However, 3.6% of patients with M0 disease died within six months, compared to 14.7% of patients with a M1 disease (p <0.001). We found that M1 patients had a 4.36-times higher risk of dying within six months compared to M0 patients (Table 3).

## Independent factors for death within six months after diagnosis

A multivariable Cox regression was carried out with death within six months as the dependent variable (Table 4). Patients diagnosed between the years 2008–2012 had a six-month mortality rate of 4.7% compared to 2.5% for patients diagnosed between 2017–2020, indicating a risk reduction of 53% (p <0.001) for death within six months (Table 4). Females had a 20% lower risk of dying within six months compared to males (p = 0.013). For every year older the patient was at diagnosis, a 4.8% (p <0.001) higher risk of dying within six months was observed.

Patients with a WHO score of 1 had approximately 2.4-times greater risk of early death compared to WHO 0 patients (p <0.001). The risk of early death among WHO 4 patients was almost 28 times higher than for WHO 0 patients (p <0.001).

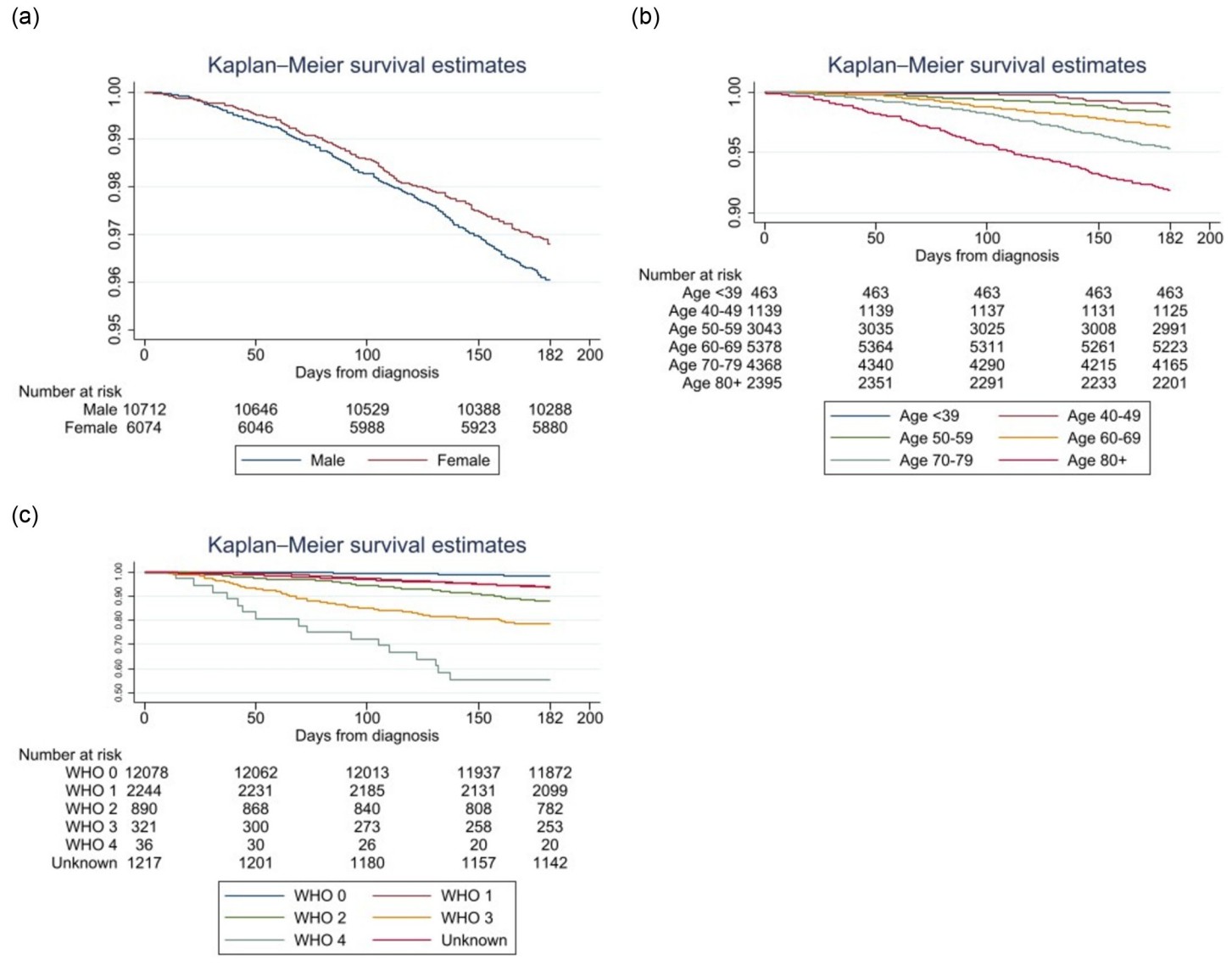

**Fig 4.** a. Kaplan–Meier curve of death within six months from diagnosis based on sex, log rank test (p = 0.012). b. Kaplan–Meier curve of death within six months from diagnosis based on age at diagnosis, log rank test (p <0.001). c. Kaplan–Meier curve of death within six months from diagnosis based on WHO function class at diagnosis, log rank test (p <0.001).

A patient with a tumor localized in the hypopharynx had a 1.58-times higher risk of dying within six months compared to a patient with a tumor in the oropharynx (p = 0.003). Patients with a T3/4 tumor had a 2.9-times higher risk of dying within six months compared to T1/2 patients (p <0.001). It was found that patients with neck metastasis at diagnosis had a 1.6-times higher risk of dying within six months compared to patients without nodal involvement (p <0.001). Distant metastasis at diagnosis gave a 2.3-times higher risk of dying within six months (p = 0.002) (Table 4).

These results form a risk profile where year of diagnosis, higher age, male sex, hypopharyngeal tumor, advanced TNM class, and higher WHO score constitute risk factors for early death.

**Table 3. Descriptive data for patients with curative treatment intent, and univariate analysis of death within 6 months of diagnosis.**

| N = 16786 | Prevalence n (%) | Dead within 6 months/N (%) | Hazard Ratio (95% CI) | P-value |
|---|---|---|---|---|
| **Tumor site** | | | | |
| Oropharynx | 4698 (28.0) | 146/4698 (3.1) | 1.00 | - |
| Oral Cavity | 4660 (27.8) | 201/4660 (4.3) | 1.40 (1.12–1.73) | 0.002 |
| Larynx | 2134 (12.7) | 89/2134 (4.2) | 1.35 (1.04–1.76) | 0.026 |
| Hypopharynx | 591 (3.5) | 61/591 (10.3) | 3.45 (2.56–4.65) | <0.001 |
| Nasopharynx | 381 (2.3) | 5/381 (1.3) | 0.42 (0.17–1.02) | 0.056 |
| Sino Nasal | 690 (4.1) | 26/690 (3.8) | 1.22 (0.80–1.85) | 0.355 |
| Salivary Gland | 1212 (7.2) | 35/1212 (2.9) | 0.93 (0.64–1.34) | 0.683 |
| Lip | 1850 (11.0) | 35/1850 (1.9) | 0.61 (0.42–0.88) | 0.008 |
| CUP | 559 (3.3) | 19/559 (3.4) | 1.10 (0.68–1.77) | 0.705 |
| Ear canal | 11 (0.1) | 1/10 (10) | 3.46 (0.48–24.7) | 0.216 |
| **Stage** | | | | |
| I | 4948 (29.5) | 73/4948 (1.5) | 1.00 | - |
| II | 2847 (17.0) | 66/2847 (2.3) | 1.58 (1.13–2.20) | 0.007 |
| III | 2077 (12.4) | 87/2077 (4.2) | 2.88 (2.11–3.93) | <0.001 |
| IV A | 5507 (32.8) | 293/5507 (5.3) | 3.68 (2.85–4.75) | <0.001 |
| IV B | 553 (3.3) | 56/553 (10.1) | 7.12 (5.03–10.1) | <0.001 |
| IV C | 93 (0.6) | 14/93 (15.1) | 11.0 (6.23–19.6) | <0.001 |
| Missing | 761 (4.5) | - | - | - |
| **T** | | | | |
| T1-2 | 11,649 (69.4) | 229/11,649 (2.0) | 1.00 | - |
| T3-4 | 4968 (29.6) | 383/4968 (7.7) | 4.04 (3.43–4.76) | <0.001 |
| Missing | 169 (1.0) | - | - | - |
| **N** | | | | |
| N- | 10,263 (61.2) | 300/10,263 (2.9) | 1.00 | - |
| N+ | 6520 (38.8) | 318/6520 (4.9) | 1.68 (1.44–1.97) | <0.001 |
| **M** | | | | |
| M0 | 16,511(98.4) | 596/16,511 (3.6) | 1.00 | - |
| M1 | 102 (0.6) | 15/102 (14.7) | 4.36 (2.61–7.28) | <0.001 |
| Missing | 173 (1.0) | - | - | - |

## Discussion

This real-world, population-based, Swedish nationwide study, including 16,786 patients with HNC and curative treatment intent demonstrated that year of diagnosis, higher age, higher TNM class, male sex, higher WHO class, and a tumor in the hypopharynx were independent risk factors for death within six months of diagnosis. This study is a unique contribution to the knowledge of early death in patients with HNC, highlighting the significant reduction of 6-month mortality of 53% from 2008 to 2020. The effects of standardized protocols to ensure that all patients receive adequate treatment without unnecessary delays might have contributed to this reduction.

Throughout the study period, patients with curative treatment intent (n = 16,786) had 3.7% six-month mortality overall. Jensen et al. reported a 7.1% six-month mortality rate in a Danish nationwide study including 11,419 HNC patients between 2000 and 2017 treated with curatively intended radio- or chemoradiotherapy. The same author studied a series of 2209 HNC patients between 2010 and 2017 from multiple centers and found a non—cancer-specific six-

**Table 4. Multivariable Cox regression with death within six months of diagnosis as target variable for HNC patients undergoing curative-intended treatment.**

| Factor | Hazard Ratio (95% CI) | Significance |
|---|---|---|
| **Year of diagnosis** | | |
| 2008–2012 | - | - |
| 2013–2016 | 0.83 (0.69–1.00) | 0.052 |
| 2017–2020 | 0.47 (0.40–0.61) | <0.001* |
| **TNM Score** | | |
| T1-2 vs 3–4 | 2.90 (2.43–3.51) | <0.001* |
| N+/0 | 1.60 (1.36–1.99) | <0.001* |
| M+/0 | 2.30 (1.36–3.85) | 0.002* |
| **Sex** | | |
| Male | - | - |
| Female | .800 (0.67–0.95) | 0.013* |
| **Age Continuously** | 1.05 (1.04–1.06) | < .001* |
| **WHO Function** | | |
| WHO 0 | - | - |
| WHO 1 | 2.38 (1.90–2.97) | <0.001* |
| WHO 2 | 3.84 (2.99–4.91) | <0.001* |
| WHO 3 | 7.87 (5.90–10.5) | <0.001* |
| WHO 4 | 27.6 (16.5–46.2) | <0.001* |
| WHO Missing | 2.90 (2.21–3.81) | <0.001* |
| **Tumor Site** | | |
| Oropharynx | - | - |
| Oral cavity | 1.21 (0.95–1.53) | 0.121 |
| Larynx | 1.13 (0.85–1.51) | 0.397 |
| Hypopharynx | 1.58 (1.16–2.14) | 0.003* |
| Nasopharynx | 0.510 (0.21–1.24) | 0.137 |
| Sino nasal | 0.984 (0.63–1.53) | 0.943 |
| Salivary Gland | 0.890 (0.61–1.31) | 0.555 |
| Lip | 0.629 (0.41–0.95) | 0.029* |
| CUP | 1.58 (0.95–2.64) | 0.077 |
| Ear | 4.84 (0.67–34.9) | 0.118 |

*significant differences in multivariable analysis

month mortality of 4.4% following treatment with radiotherapy [24, 25]. Nieminen et al. reported a six-month mortality of 11.4% in a subgroup of 317 HNC patients undergoing microvascular reconstruction [26]. As patients with HNC comprise a heterogenous population, the early mortality rate differs if subpopulations of patients are analyzed. Studies looking merely at treatment with radiotherapy, or at microvascular reconstruction contain pre-selected patient cohorts likely affecting the rate of early death. Our study includes all surgical and oncological treatment options for patients with curative treatment intent and all T classes. Patients undergoing treatment with a free flap and microvascular reconstruction typically have at least a T2 tumor [17], indicating a disease with a higher risk of early death.

In the present study, men had a higher risk of early death, in accordance with the findings by Kouka et al. in their series of 8288 German HNC patients [27]. Other studies have not shown a sex-dependent rate of early death in HNC patients [12, 24]. In this study there is an uneven distribution between sexes regarding tumor stage and site; for example, only 23.5% of

hypopharyngeal cancer patients were female, which can partly explain the better prognostic outcome for females.

Other studies have suggested that estrogen could have a beneficial impact on survival after HNC [28–31], however that correlation was not analyzed here. In consistency with other studies [8, 12, 24, 26, 27, 32, 33], we saw that higher age and performance status correlated with an increased rate of early death. Furthermore, a hypopharyngeal tumor location, higher stage, and advanced T class were confirmed as significant independent risk factors for six-month mortality, in accordance with previous findings [12, 32, 34, 35].

Our study has revealed that the six-month mortality decreased throughout the 2008–2020 period. The reasons are most likely multifactorial. Sweden implemented standardized diagnostic work ups in 2015, which led to faster diagnoses and shortened waiting times before treatment for HNC patients. The same year, national guidelines for HNC treatment were published, leading to a national standardized consensus regarding diagnostic work-up and treatment [18]. Furthermore, oropharyngeal cancer derived from HPV is increasing [2, 7, 36–38], and has a better prognosis compared to HPV-negative tumors [39, 40]. The substantial increase in HPV-positive cancers and the decreased prevalence of daily smokers in Sweden (12% in 2008 to 8% in 2020), contributing to less HPV-negative HNC cases can be part of the explanation as to why six-month mortality is decreasing for HNC patients [41–43].

It is reasonable to speculate that a number of variables not registered in SweHNCR have improved the diagnostic work-up, treatment, and care of patients with HNC. Neck dissection was implemented 2015 in Sweden as the standard treatment for oral cavity cancer patients with a depth of invasion of the primary tumor exceeding 3 millimeters and N0 status [18]. This change is likely contributing to more active treatment of occult metastasis, and therefore to improved survival rates. It is also likely that novel innovative therapy has had an impact on early mortality. Since 2008, there have been several improvements in oncological treatments such as a change in external radiotherapy from using 3D conformal radiotherapy to intensity-modulated radiotherapy and volumetric modulated arc radiotherapy, shifts from neoadjuvant high-dose cisplatin regimens to concomitant cisplatin. Concomitant cetuximab treatment was proven to be less effective than concomitant cisplatin, and was excluded from routine treatment [44]. The rise of checkpoint inhibitors in the treatment of cancer have altered the prognosis for many different types of cancers [45]. However, it should be pointed out that cancer immunotherapy is only used in patients receiving palliative treatment for HNC [17]. New diagnostic tools have been increasingly available for accurate HNC diagnosis. The implementation of sentinel node biopsies in Sweden could also have contributed to identifying more contra- or bilateral metastases [46]. Moreover, the use of Positron Emission Tomography with 2-deoxy-2-[fluorine-18] fluoro- D-glucose integrated with Computed Tomography, FDG-PET/CT has improved the detection of regional and distant metastases [47, 48]. Altogether, this study suggests that standardized and rapid diagnostic work-up and treatment, together with other factors, contributes to fewer HNC patients dying within six months of diagnosis.

The limitations of the study includes its retrospective setting [49], the fact that important parameters influencing overall survival, such as comorbidities, alcohol consumption, HPV status, and smoking habits were not recorded in the early years of SweHNCR. Actual cause of death was not reported in the register, nor were data on occupational exposures, socioeconomic factors, oral hygiene, complications to treatments, and gene mutations; all factors which could influence early mortality [1, 33, 50, 51].

Compensating for several of the study's limitations is the large population of 16,786 consecutive HNC patients with curative treatment intent, and the homogeneity of healthcare provided to all Swedes. The results in this study can be used to identify patients at high risk of

early death, and thus to improve therapeutical decision making and more personalized HNC management.

## Conclusion

This first European population-based analysis of trends in six-month mortality among HNC patients with curative treatment intent is based on registry data comprising 16,786 patients. The study provides generalizable epidemiological data on early mortality and survival of curative HNC patients, in a single-payer funded healthcare setting. The mean time to treatment was significantly decreased between the years 2008 and 2012 compared to 2017–2020. Encouragingly, the present results show that early mortality has decreased from 4.7% (2008–2012) to 2.5% (2017–2020) in Sweden. Female patients had a 20% lower risk of death within six months compared to male patients.

Increased age at diagnosis, male sex, higher WHO function class, advanced TNM class, and tumor in the hypopharynx were independent risk factors for death within six months.

## Acknowledgments

We would like to thank Mats Fredriksson, statistician at Linköping University, for excellent statistical guidance.

## Author Contributions

**Conceptualization:** Charbél Talani, Göran Laurell, Antti Mäkitie, Lovisa Farnebo.

**Data curation:** Charbél Talani, Anders Högmo, Lovisa Farnebo.

**Formal analysis:** Charbél Talani, Anders Högmo, Lovisa Farnebo.

**Funding acquisition:** Charbél Talani, Lovisa Farnebo.

**Investigation:** Charbél Talani, Lovisa Farnebo.

**Methodology:** Charbél Talani, Göran Laurell, Antti Mäkitie, Lovisa Farnebo.

**Project administration:** Charbél Talani, Anders Högmo, Lovisa Farnebo.

**Resources:** Charbél Talani, Göran Laurell, Lovisa Farnebo.

**Software:** Charbél Talani, Lovisa Farnebo.

**Supervision:** Charbél Talani, Anders Högmo, Antti Mäkitie, Lovisa Farnebo.

**Validation:** Charbél Talani, Göran Laurell, Antti Mäkitie, Lovisa Farnebo.

**Visualization:** Charbél Talani, Lovisa Farnebo.

**Writing – original draft:** Charbél Talani, Anders Högmo, Antti Mäkitie, Lovisa Farnebo.

**Writing – review & editing:** Charbél Talani, Anders Högmo, Göran Laurell, Antti Mäkitie, Lovisa Farnebo.

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
