## [Decision Letter · Decision Letter 0]

15 Jan 2024

PONE-D-23-41780Six-month mortality has decreased for patients with curative treatment for head and neck cancer in SwedenPLOS ONE

Dear Dr. Talani,

Thank you for submitting your manuscript to PLOS ONE. After careful consideration, we feel that it has merit but does not fully meet PLOS ONE’s publication criteria as it currently stands. Therefore, we invite you to submit a revised version of the manuscript that addresses the points raised during the review process.

We look forward to receiving your revised manuscript.

Kind regards,

Chung-Ta Chang

Academic Editor

PLOS ONE

Journal Requirements:

3. Thank you for stating the following financial disclosure:"The Östergötland Regional Research Council, SC-2018-00231-53, SC-2018-00231-46"

4. We note that your Data Availability Statement is currently as follows: "All relevant data are within the manuscript and its Supporting Information files."

Reviewers' comments:

Reviewer's Responses to Questions

**Comments to the Author**

1. Is the manuscript technically sound, and do the data support the conclusions?

Reviewer #1: Yes

Reviewer #2: Yes

2. Has the statistical analysis been performed appropriately and rigorously? 

Reviewer #1: Yes

Reviewer #2: Yes

3. Have the authors made all data underlying the findings in their manuscript fully available?

Reviewer #1: No

Reviewer #2: Yes

4. Is the manuscript presented in an intelligible fashion and written in standard English?

Reviewer #1: No

Reviewer #2: Yes

5. Review Comments to the Author

Reviewer #1: This retrospective cohort study of patients in Sweden with HNC analyzed the incidence of different HNC cancers and the trends in early death during the time period of 2008-2020. The study results are important and deserve to be published after major revision of the article.

Strengths include the large database contributing substantial N for analyses and, I assume, the amount of clinical detail available on the patients. The ethical statements are sufficient. I found the decline in 6-month mortality interesting and encouraging. However, it is important to note that although there was a statistically significant decline in 6-month mortality post-diagnosis, overall, it remained low for curative approach treatment for HNC across all periods.

The statistical and analytic methods appear appropriate but are not well presented in the Methods, Tables, Results, or Figures. This hinders both the ability to review the methods and any future efforts to replicate the findings. For example, there should be baseline characteristics tables for each cohort (at least in the Supplemental), separate from the analytic results tables, so that the populations can be fully understood. The distribution of treatments received over time across cohorts should be reported. All Kaplan-Meier charts should contain risk tables underneath them per best reporting standards.

In general, there is low precision in the writing that needs to be improved, including missing citations and use of vague language. I noted numerous factual statements lacking a citation and insufficient explanations of statements for a general, not explicitly clinical audience of this journal. The Methods section in both the abstract and the main article are missing essential details, including a clear statement of the outcomes assessed, the comparisons performed, and the explicit inclusion/exclusion criteria. All outcomes and comparisons need to be fully described in the Methods. The separation of the cohorts by the years chosen needs to be rationalized.

Clarity and organization overall need improvement and the article would benefit from English language editing for grammar and flow. The authors are encouraged to include subheadings in the Results that improve the ease of reading, beginning with 'Patient Characteristics' and then one for each outcome assessed. All tables and figures should include appropriate captions describing the comparison displayed, defining abbreviations, notes, etc.

Previous literature on the topic of early death in HNC should be mentioned in the Introduction with better precision in terms of years, outcomes, regional populations, trends over defined periods, etc. The current treatment guidelines for HNC in Sweden should be briefly mentioned in the Introduction, as well as the gap in knowledge leading to the rationale for this study. The Discussion needs to highlight the unique contributions of this study in comparison with prior findings, which is currently not clear.

Throughout, it should be clarified that this is a study in a population of Swedish patients (vs. European or HNC patients generally). This is important as Sweden has a single-payer healthcare system and likely different available treatments for HNC compared to other nations. The impact of the advent of targeted or immunotherapy for HNC in Sweden should be discussed in the context of improved survival. The generalizability of the study findings to other regions and HNC population as a whole should be addressed in the Discussion. The limitations section should be more comprehensive and include the general limitations of retrospective cohort studies.

Reviewer #2: The authors aimed to identify risk factors for early death among patients with curative treatment, and to analyze whether the risk of early death changed over the years. In general, this is a well-written article. However, there are still some points need to be addressed.

1. In the Statistics section, the authors mentioned “… and Fisher’s exact test was used for tables with dichotomous variables (Sex, TNM score, and death).” Why these three variables were tested by Fisher’s exact test but not chi-square test?

2. In the Statistics section, the authors mentioned “Exact binomial confidence intervals were estimated forproportions. …..” Please correct “forproportions” to “for proportions.”

3. In Table 1, the authors had shown “Age continuous (mean(sd)) was 66.1 (12.6). Please delete the redundant word of ” continuous”.

4. In Table 2, the “Total patients” column is redundant and the numbers are wrong. Please correct or delete this column. Moreover, the authors should also show the “Stage” as variable to see if any difference in these three groups.

5. In the “Independent factors for death within six months after diagnosis” section, the authors mentioned “A 67-year-old patient had a 4.8% higher risk of dying within six months than a patient who was 66 years old.” Please revised this sentence to correct sentence.

6. In Table 4, the authors should mention that it is calculated by male vs female or female vs male in factor “Sex”. Moreover, the authors should also include “Stage” as factor in multivariable Cox regression analysis since this factor is also significantly different in univariate analysis.

7. In Figure 3 and 4, the authors should also mark the p values in these figures.

6. PLOS authors have the option to publish the peer review history of their article (what does this mean?). If published, this will include your full peer review and any attached files.

Reviewer #1: No

Reviewer #2: No

---

## [Author Response · Author response to Decision Letter 0]

20 Feb 2024

The manuscript meets PLOS ONE´s style requirements.

Thank you for highlighting this, the Funding Information section has now been revised and a financial disclosure section has been added as requested.

3. Thank you for stating the following financial disclosure:"The Östergötland Regional Research Council, SC-2018-00231-53, SC-2018-00231-46"

Thank you for this input. It has been added in the new file named ‘Financial disclosure’ and uploaded in the revised submission.

The following statement has been added to the cover letter:

“The authors received funding for the work with this article but funders had no role in study design, data collection and analysis, decision to publish, or preparation of the manuscript.”

4. We note that your Data Availability Statement is currently as follows: "All relevant data are within the manuscript and its Supporting Information files."

Data is owned by a third party, the Regional Cancer Center Väst, in Sweden, and the data set even though de-identified can therefore not be shared. Data can be accessed through a formal application to the third-party organization via the following link: https://cancercentrum.se/samverkan/vara-uppdrag/kunskapsstyrning/kvalitetsregister/datauttag/

Reviewers' comments:

Reviewer's Responses to Questions

Comments to the Author

1. Is the manuscript technically sound, and do the data support the conclusions?

Reviewer #1: Yes

Reviewer #2: Yes

2. Has the statistical analysis been performed appropriately and rigorously? 

Reviewer #1: Yes

Reviewer #2: Yes

3. Have the authors made all data underlying the findings in their manuscript fully available?

Reviewer #1: No

We wish to thank the Reviewers for reviewing our work so thoroughly. Revisions have been made throughout the manuscript and appear highlighted in the revised paper.

Thank you for highlighting the issue of raw data. Data are owned by a third party, the Regional Cancer Center Väst and can be accessed through a formal application to them via the following link: https://cancercentrum.se/samverkan/vara-uppdrag/kunskapsstyrning/kvalitetsregister/datauttag/

Reviewer #2: Yes

4. Is the manuscript presented in an intelligible fashion and written in standard English?

Reviewer #1: No

Revisions have been made as requested and can be visualized in the document with tracked chances. 

We would like to point out that a language revision has been carried out twice, first by CBG consult before submission in November 2023 and also after after our revisions in February 2024. 

Reviewer #2: Yes

5. Review Comments to the Author

Reviewer #1: This retrospective cohort study of patients in Sweden with HNC analyzed the incidence of different HNC cancers and the trends in early death during the time period of 2008-2020. The study results are important and deserve to be published after major revision of the article.

Strengths include the large database contributing substantial N for analyses and, I assume, the amount of clinical detail available on the patients. The ethical statements are sufficient. I found the decline in 6-month mortality interesting and encouraging. However, it is important to note that although there was a statistically significant decline in 6-month mortality post-diagnosis, overall, it remained low for curative approach treatment for HNC across all periods. The statistical and analytic methods appear appropriate but are not well presented in the Methods, Tables, Results, or Figures. This hinders both the ability to review the methods and any future efforts to replicate the findings. 

For example, there should be baseline characteristics tables for each cohort (at least in the Supplemental), separate from the analytic results tables, so that the populations can be fully understood. 

Thank you for the excellent remarks. 

Table 2 has been extended to now contain age group distribution between the three different cohorts, highlighted in Table 2.

The distribution of treatments received over time across cohorts should be reported. 

Table 2 has been expanded to include the distribution of treatment in the three cohorts.

All Kaplan-Meier charts should contain risk tables underneath them per best reporting standards.

Thank you for this clarifying remark. Risk tables have now been added to the Kaplan-Meier charts in Fig 3 and 4a-c.

In general, there is low precision in the writing that needs to be improved, including missing citations and use of vague language. I noted numerous factual statements lacking a citation and insufficient explanations of statements for a general, not explicitly clinical audience of this journal. 

Thank you for pointing this out. These issues have been addressed throughout the manuscript and changes are marked in yellow. References have been added as requested by the Reviewer 1 and language improvements have been made for clarification accordingly.

The Methods section in both the abstract and the main article are missing essential details, including a clear statement of the outcomes assessed, the comparisons performed, and the explicit inclusion/exclusion criteria. 

Thank you for addressing this matter. In our nation-wide study we excluded all patients below the age of 18 years. This exclusion criterion was added to the Abstract and the main article. All patients above the age of 18 with a head and neck cancer diagnosis were included. Merely data for 1,5% of the patients were missing. Unfortunately, it is not known why these patients were missing or who they were, but it can be speculated that they were patients having for example T1 lip cancer, and thus having been surgically treated by another specialist than an ENT-doctor/Head and Neck surgeon. Other specialists are not as well informed about the registration of head and neck cancers in the SweHNCR and can thereby miss to register their case. Palliative patients were not included in the statistical analysis since this paper focused on patients with curative treatment intent, as shown in Fig 1. 

Fig 1 has clarified this by adding colored boxes in the flow chart and addressing the inclusion of patients with curative treatment intent only. This is now also described in the Figure Legends. 

All outcomes and comparisons need to be fully described in the Methods. The separation of the cohorts by the years chosen needs to be rationalized.

This has been addressed in Table 2, as described above. A rationale for the separation of the cohort has been added to the methods section. This separation with 5, 4, and 4 years in the groups seems logical since a 24% incidence increase in HNC during the time period was noted.

Clarity and organization overall need improvement and the article would benefit from English language editing for grammar and flow. 

As stated above, English language revision was carried out before submission and again before re-submission.

The authors are encouraged to include subheadings in the Results that improve the ease of reading, beginning with 'Patient Characteristics' and then one for each outcome assessed. 

For clarity reasons all headings have been reviewed and subheadings were added on page 4 “Data Source”, and page 6 “Epidemiology”. Style and font size of headings were corrected to add clarification to the subsections.

All tables and figures should include appropriate captions describing the comparison displayed, defining abbreviations, notes, etc.

All tables and figures were revised according to the instructions given by Reviewer 1.

Previous literature on the topic of early death in HNC should be mentioned in the Introduction with better precision in terms of years, outcomes, regional populations, trends over defined periods, etc. 

Population-based studies on early death are scarce, which is why we wish to address this subject in our large, nationwide population. We wish to thank the Reviewers for revising our work so thoroughly. Revisions have been made throughout the manuscript and the references on early death defined as, death within six months in HNC are added in the second paragraph on page 3. 

The current treatment guidelines for HNC in Sweden should be briefly mentioned in the Introduction, as well as the gap in knowledge leading to the rationale for this study. 

Thank you for pointing out this important view. The Introduction has been revised according to the Reviewers’ comments, and highlighted in the manuscript on page 3. 

The Discussion needs to highlight the unique contributions of this study in comparison with prior findings, which is currently not clear.

The unique contribution of this study is now emphasized in the Discussion subsection on page 15.

Throughout, it should be clarified that this is a study in a population of Swedish patients (vs. European or HNC patients generally). This is important as Sweden has a single-payer healthcare system and likely different available treatments for HNC compared to other nations. The impact of the advent of targeted or immunotherapy for HNC in Sweden should be discussed in the context of improved survival.

Both the Introduction and the Discussion sections have been revised to underscore that the cohort is Swedish. 

Targeted therapy/ immunotherapy in Sweden is currently only used in the palliative setting. This study excludes palliative patients, emphasized in the Methods, and Discussions sections. 

The generalizability of the study findings to other regions and HNC population as a whole should be addressed in the Discussion. 

Thank you – a discussion on the generalizability has been added in the last sentence of the Discussion section, on page 18.

The limitations section should be more comprehensive and include the general limitations of retrospective cohort studies.

We added the limitation of a retrospective study design according to the Reviewers’ request on page 18, including a reference elaborating on the weaknesses of a retrospective setting (Talari et al).

Reviewer #2: The authors aimed to identify risk factors for early death among patients with curative treatment, and to analyze whether the risk of early death changed over the years. In general, this is a well-written article. However, there are still some points need to be addressed.

1. In the Statistics section, the authors mentioned “… and Fisher’s exact test was used for tables with dichotomous variables (Sex, TNM score, and death).” Why these three variables were tested by Fisher’s exact test but not chi-square test?

Sex, TNM score, and death were analyzed using Chi-square test, and the Statistics section was revised according to the Reviewers’ suggestion.

2. In the Statistics section, the authors mentioned “Exact binomial confidence intervals were estimated for proportions. …..” Please correct “forproportions” to “for proportions.”

This has been corrected, thank you for pointing it out. 

3. In Table 1, the authors had shown “Age continuous (mean(sd)) was 66.1 (12.6). Please delete the redundant word of ” continuous”.

This has been corrected, thank you for pointing it out.

4. In Table 2, the “Total patients” column is redundant and the numbers are wrong. Please correct or delete this column. Moreover, the authors should also show the “Stage” as variable to see if any difference in these three groups.

Thank you for this valuable comment. The “Total patients” column has been deleted and Table 2 was revised and now contains data on stage, Age, and Treatment.

5. In the “Independent factors for death within six months after diagnosis” section, the authors mentioned “A 67-year-old patient had a 4.8% higher risk of dying within six months than a patient who was 66 years old.” Please revised this sentence to correct sentence.

We agree with the Reviewer 2 that this sentence was redundant and have now removed it from the manuscript.

A 67-year-old patient had a 4.8% higher risk of dying within six months than a patient who was 66 years old

6. In Table 4, the authors should mention that it is calculated by male vs female or female vs male in factor “Sex”. 

Thank you for pointing out this mistake.

The calculation is performed as female vs male, and male having hazard ratio set to 1. This has been corrected in Table 4 by adding two new rows including male and female, and is now highlighted in the revised manuscript.

Moreover, the authors should also include “Stage” as factor in multivariable Cox regression analysis since this factor is also significantly different in univariate analysis.

TNM and stage are overlapping factors in a multivariable analysis, since they both describe the severity of the disease. Furthermore, stage is derived from the TNM classification. We chose TNM classification over stage in the multivariable analysis, because TNM gives a more detailed risk stratification in the multivariable analysis. We wanted to keep the possibility of showing that an advanced T-class had a higher hazard ratio for death within 6 months than regional metastasis, something that stage could not discriminate.

7. In Figure 3 and 4, the authors should also mark the p values in these figures.

P-values have been added to the figure legends in Figures 3 and 4a-c.

Figure 2 has also been revised according to Reviewers’ remarks. Axis legend has been enlarged and numerical data were added in Fig 2a.

---

## [Editor Report · Decision Letter 1]

26 Feb 2024

Six-month mortality has decreased for patients with curative treatment intent for head and neck cancer in Sweden

PONE-D-23-41780R1

Dear Dr. Talani,

We’re pleased to inform you that your manuscript has been judged scientifically suitable for publication and will be formally accepted for publication once it meets all outstanding technical requirements.

Kind regards,

Chung-Ta Chang

Academic Editor

PLOS ONE
---

## [Editor Report · Acceptance letter]

2 Apr 2024

PONE-D-23-41780R1 

PLOS ONE

Dear Dr. Talani, 

I'm pleased to inform you that your manuscript has been deemed suitable for publication in PLOS ONE. Congratulations! Your manuscript is now being handed over to our production team.

Kind regards, 

on behalf of

Dr. Chung-Ta Chang 

Academic Editor

PLOS ONE